# The Gut Microbial Regulation of Epigenetic Modification from a Metabolic Perspective

**DOI:** 10.3390/ijms25137175

**Published:** 2024-06-29

**Authors:** Xingtong Lin, Hui Han, Nan Wang, Chengming Wang, Ming Qi, Jing Wang, Gang Liu

**Affiliations:** 1College of Animal Science and Technology, Hunan Agricultural University, Changsha 410128, China; lin08232021@163.com (X.L.); hanhui16@mails.ucas.ac.cn (H.H.); wangnan0317@stu.hunau.edu.cn (N.W.); wangcm1028@stu.hunau.edu.cn (C.W.); qmcharisma@sina.com (M.Q.); 2Yuelushan Laboratory, Changsha 410128, China; 3College of Bioscience and Biotechnology, Hunan Agricultural University, Changsha 410128, China

**Keywords:** obesity, gut microbiota, metabolites, epigenetic modification, therapy

## Abstract

Obesity is a global health challenge that has received increasing attention in contemporary research. The gut microbiota has been implicated in the development of obesity, primarily through its involvement in regulating various host metabolic processes. Recent research suggests that epigenetic modifications may serve as crucial pathways through which the gut microbiota and its metabolites contribute to the pathogenesis of obesity and other metabolic disorders. Hence, understanding the interplay between gut microbiota and epigenetic mechanisms is crucial for elucidating the impact of obesity on the host. This review primarily focuses on the understanding of the relationship between the gut microbiota and its metabolites with epigenetic mechanisms in several obesity-related pathogenic mechanisms, including energy dysregulation, metabolic inflammation, and maternal inheritance. These findings could serve as novel therapeutic targets for probiotics, prebiotics, and fecal microbiota transplantation tools in treating metabolic disruptions. It may also aid in developing therapeutic strategies that modulate the gut microbiota, thereby regulating the metabolic characteristics of obesity.

## 1. Introduction

The issue of obesity is constantly on the rise globally. The global prevalence of obesity continues to rise at an alarming rate [1]. Recent statistics reveal that over 200 million adults are afflicted with obesity or other metabolic diseases, accounting for approximately 30% of the world’s population [2]. Obesity has been identified as a global health challenge as it increases the probability of developing various chronic conditions, including type 2 diabetes, cardiovascular diseases, and premature aging [3,4,5]. The pathological physiology and etiology of obesity involve multiple factors, including environmental factors, an imbalance between energy intake and expenditure, immune response, and genetic factors. Growing evidence demonstrates that epigenetic modifications are one of the mechanisms linking altered gene activity to environmental factors that contribute to the occurrence and development of obesity [6]. Epigenetic modifications are heritable changes in gene function during mitosis or meiosis without alterations in DNA sequence, including DNA methylation, histone modification, chromatin remodeling, and regulation by non-coding RNAs [7]. Epigenetic processes regulate the expression of numerous genes, including genes involved in metabolism and inflammation pathways [8,9]. Recent studies have demonstrated the distinct epigenetic signatures in subjects with obesity [10], which are potential biomarkers of obesity and metabolic disease risk. Therefore, a comprehensive understanding of the underlying epigenetic mechanisms involved in the development of obesity is crucial. Such knowledge has the potential to pave the way for promising therapeutic strategies to combat obesity.

The gut microbiota has been recently demonstrated as a key environmental factor in the development of obesity and its related diseases [11,12]. A large body of evidence from animal and human studies indicates changes in the gut microbiota composition and function in obese individuals [13,14,15]. For example, individuals with obesity have lower microbial α-diversity [16] and higher levels of microbiota that exhibited increased energy-harvesting ability [17,18]. Epigenetic modifications play an important role in the relationship between the gut microbiota and obesity development. Evidence has indicated that the gut microbiota and its metabolites can directly influence epigenetic pathways by regulating host-cell intrinsic processes or generating epigenetic substrates and enzymatic cofactors to influence the host’s metabolism [19,20,21]. Therefore, unraveling the potential mechanisms of the crosstalk between the gut microbiota and epigenetic modifications is vital for understanding the development of obesity.

In this review, we examine the role of the gut microbiota and its metabolites as epigenetic modifiers in the development of obesity and thoroughly describe the potential mechanisms by which the gut microbiota modulates obesity by mediating epigenetic modulations. Previous studies have reported the association between the gut microbiota and epigenetic modulations in the development of obesity and associated comorbidities [22,23,24]. In this review, we provide an updated focus on recent evidence regarding the interaction between the gut microbiota and epigenetic mechanisms, specifically in the regulation of gene expression profiles and phenotypic outcomes in obesity from the perspective of energy metabolism, metabolic inflammation, and maternal inheritance. Additionally, we extensively discuss the impact of the so-called chemical crosstalk between microbial metabolites and associated epigenetic modifications on the development of obesity. We also explore novel microbiome-targeted therapies for the treatment of obesity through epigenetic mechanisms, such as probiotics, prebiotics, and fecal microbiota transplantation. These scientific insights will provide a theoretical basis for the potential use of the gut microbiota as a strategy for the management of obesity.

## 2. Literature Search Methodology

Searching PubMed, Web of Science, and Google databases for human and animal in vivo studies/clinical trials focused on gut microbial–epigenetic modification in obesity. The search was restricted to English-language studies regarding publication dates from 2006 to January 2024. We also included two important articles published in 1978 and 1997. The search terms included the following: “gut microbiota”, “gut microbiome”, “epigenetics”, “epigenetic regulation”, “obesity”, and “overweight”. Some research, such as studies conducted on non-obese hosts, was excluded from the review.

## 3. Epigenetic Regulation Linking the Gut Microbiota and Obesity

Epigenetics refers to modifications in chromatin structure and function that do not involve alterations in the underlying DNA sequence. These changes encompass various processes, including DNA methylation, the modifications of histones, and mechanisms mediated by RNA [25,26]. Mounting evidence suggests that the gut microbiota can influence host epigenetic regulation, thereby impacting the onset and progression of obesity [27,28]. Gaining a deeper understanding of the epigenetic link between the gut microbiota and obesity could present opportunities to reduce the incidence and consequences of obesity. Figure 1 provides an overview of the interplay between the gut microbiota, epigenetic modifications, and obesity-related diseases. We will hereafter summarize the effects of changes in the gut microbiota on epigenetic regulation, which modulates the development of obesity via regulating energy metabolism, inflammatory response, and genetic factors.

### 3.1. Gut Microbiota–Epigenetic Modification in Energy Metabolism

The maintenance of systemic energy homeostasis relies mainly on the balance between energy intake and expenditure. When energy intake surpasses expenditure, an imbalance occurs in the systemic energy homeostasis, leading to the accumulation of adipose tissue volume and quantity, which ultimately results in obesity [29,30,31,32]. It has been reported that the gut microbiota can impact host metabolism by inducing epigenetic alterations in key genes involved in regulating energy metabolism [33,34,35]. Hence, the regulatory role of the interplay between the gut microbiota and epigenetic modifications in energy metabolism is increasingly being investigated within the context of obesity.

Non-coding RNAs (ncRNAs) are functional RNA molecules present in the genome that do not encode proteins. MicroRNAs (miRNAs), as evolutionarily conserved short non-coding RNA molecules, primarily participate in the regulation of gene expression and protein translation [36]. Currently, there is a growing interest in understanding the role of miRNAs in obesity and related metabolic disorders by influencing the biology (development and metabolism) of adipose tissue [37]. A previous study using germ-free (GF) mice showed that the gut microbiota is causal in controlling adipocyte miR-181 expression to regulate glucose and energy homeostasis during obesity [38]. A recent review has also thoroughly discussed the association between gut dysbiosis and miRNA in metabolic disorders [39]. This article proposes that the gut microbiota affects host metabolism mainly through lipopolysaccharide and secondary microbial metabolites regulating host microRNA. Therefore, this confirms the possibility of the gut microbiota–miRNA axis as a new target for treating metabolic disorders in obesity.

Histone modifications typically do not directly target DNA but covalently add lysine (K) residues to the histone tails. The main modifications include histone acetylation and deacetylation [40]. HDACs have been demonstrated to function as critical regulatory factors involved in lipid and other metabolic pathways [41]. Kuang et al. discovered that the gut microbiota controls lipid metabolism through HDAC3 in the mouse intestine, leading to increased expression of the lipid transporter CD36 and promoting lipid uptake by intestinal epithelial cells, thereby exacerbating the development of obesity [42]. This research establishes the relationship between the histone deacetylase family and the gut microbiota in the regulation of energy lipid metabolism.

DNA methylation, a crucial epigenetic mechanism, regulates gene expression by adding methyl groups to DNA molecules [43,44]. An increasing number of perspectives indicate that different microbial characteristics of obese individuals may trigger changes in DNA methylation patterns. For instance, Ramos-Molina et al. have found that the relative abundance of *Bacteroidetes* in obese patients was positively correlated with the methylation levels of the promoter regions of *HDAC7* gene (*p* = 0.011) and insulin-like growth factor 2 mRNA-binding protein 2 gene (*IGF2BP2*) (*p* = 0.002) in adipose tissue. In contrast, the relative abundance of *Firmicutes* was negatively correlated with the methylation level of the promoter region of *HDAC7* in blood (*p* = 0.019) [45]. A clinical study showed that the obese subjects with a high *Bacteroidetes*-to-*Firmicutes* ratio exhibited different DNA methylation patterns in the blood and adipose tissue when compared with those in the obese subjects with a low *Bacteroidetes*-to-*Firmicutes* ratio [46]. It has been reported that insulin and leptin signaling play a critical role in modulating glucose and lipid metabolism, and thus contribute to the development of obesity [47,48,49]. Salas-Perez et al. established a connection between the gut microbiota and DNA methylation in individuals with obesity [50], specifically noting that the effect of Ruminococcus abundance on BMI was mediated by the methylation of the macro domain containing 2 gene (*MACROD2*) and differentially methylated region gene (*DMR*) (*p* = 0.035). Additionally, compared to conventional mice, GF mice exhibited an increase in the DNA methylation of the leptin promoter CpG (cytosine-guanine dinucleotide) of adipose tissue by approximately 6% to 16% (*p* < 0.05), which might indicate an increased risk factor of leptin resistance [51]. Kumar H. et al. investigated a significant association between bacterial dominance and epigenetic profiles in eight pregnant women. The results indicated that obese pregnant women had a gut microbiota dominated by the Firmicutes phylum and exhibited a higher degree of methylation in the promoter region of the Stearoyl-CoA desaturase 5 gene (*SCD5*) [46]. These findings confirm that the crosstalk between the gut microbiota and energy-metabolism-related genes can be achieved through epigenetic mechanisms.

### 3.2. Gut Microbiota–Epigenetic Modification in Low-Grade Inflammation

Obesity is frequently accompanied by various chronic complications, leading to the activation of cytokines and inflammation-related signaling pathways [52,53]. Growing evidence supports the significant role of the gut microbiota in the epigenomic remodeling of inflammatory factors [54,55].

Studies have found that changes in the gut microbiota directly influence the epigenetic modifications of TLR-mediated inflammatory molecules through DNA methylation. Specifically, Remely et al. demonstrated that obese individuals with a higher ratio of *Firmicutes*/*Bacteroidetes* showed reduced DNA methylation levels in the promoter region of the toll-like receptor 4 gene *(TLR4*) (*p* < 0.05) [56]. In addition to DNA methylation, miRNAs also play a crucial role in the inflammatory response and participate in the differentiation and function of various immune cells [57]. In a recent study, the impact of miRNA-29a on gut microbiota composition and inflammatory response in mice fed with a high-fat diet (HFD) was investigated. The results showed that compared to wild-type (WT) mice, miR-29a-overexpression mice were able to improve lipid metabolism disorders induced by a high-fat diet and promote the enrichment of *Lactobacillus* (*p* = 0.034), *Ruminiclostridium_9* (*p* < 0.001), and *Lachnoclostridium* (*p* < 0.001) in the intestine. Furthermore, it significantly reduced the expression of interleukin-6 gene (*IL-6*) in the intestine (*p* < 0.05) [58,59]. These studies confirm that epigenetic modifications can serve as a means to influence gut microbiota-host metabolic interactions and the inflammatory state induced by obesity. Additionally, several studies have indicated that changes in the gut microbiota in obesity are closely linked to the role of epigenetics in low-grade inflammation, as shown in Table 1. In general, the intestinal microbiota’s relative abundance is higher than the fecal microbiota, which may be related to the dynamic and heterogeneous nature of the microbiota along the intestinal tract [60]. However, sampling the human intestine without disturbance or contamination has always been challenging. Therefore, the human trials in Table 1 primarily use feces as the main source of information for studying the human gut microbiome [61].

Overall, there is a bidirectional relationship between the gut microbiota and epigenetic modifications of inflammatory molecules in obesity. As a result, this intricate interplay is increasingly acknowledged as a novel therapeutic and preventive approach to combat obesity. However, further research is still required to elucidate the feasibility of implementing this method.

### 3.3. Gut Microbiota–Epigenetic Modification in Maternal Inheritance

Obesity, as a multifactorial disease, is widely recognized as a major risk factor influencing the health of both children and adults [67,68]. Increasing research suggests that maternal nutrition and gut microbiota composition during pregnancy are major factors stimulating epigenetic modifications of genes related to obesity susceptibility in the fetus [69,70].

As highlighted in Table 2, an analysis based on dominant bacterial phyla in pregnant women revealed the methylation levels at the CpG sites of the ubiquitin-conjugating enzyme E2 D2 (*UBE2E2*) (*p =* 0.04) and potassium voltage-gated channel subfamily Q member 1 (*KCNQ1*) (*p* = 0.048) were positively correlated with the abundance of the maternal intestinal Firmicutes [71]. In line with this, the supplementation of probiotics in obese pregnant women altered the composition of the gut microbiota and led to reduced DNA methylation levels in the promoter regions of insulin-like growth-factor-binding protein 1 gene (*IGFBP1*) (*p* < 0.001) in their offspring [72]. This suggests that the gut microbiota confers health benefits to children by reducing the risk of glucose metabolism disorders. Experimental data from obese pregnant mice indicate that maternal obesity results in decreased microbial diversity in the cecum of the offspring, as well as alterations in the methylation patterns in DMRs of genes associated with fat metabolism, such as PPARG coactivator 1 β (*Ppargc1β*), fibroblast growth factor *(Fgf21*), EPH receptor B2 (*Ephb2*), and Von Willebrand Factor (*VWF*) (*p* < 0.05) [73]. Moreover, the offspring of pregnant mice fed a high-fat diet exhibited significantly reduced DNA methylation of cyclin-dependent kinase inhibitor 1A (Cdkn1a) in the liver, accompanied by alterations in the gut microbiota profile [74,75]. This research demonstrates the lasting impact of metabolic dysregulation induced by maternal obesity on the health of offspring, including dysbiosis in the gut microbiota and changes in the DNA methylation patterns of related genes.

Overall, the interaction between gut microbes and host epigenetics plays a multifaceted role in the mechanisms of obesity development. However, the crosstalk between the gut microbiota, epigenetics, and obesity holds potential biomedical significance and requires confirmatory evidence from more rigorous testing in clinical trials.

## 4. The Crosstalk between Gut Microbial Metabolites and Epigenetic Modification in Obesity

For a considerable period, gut microbial metabolites have been considered to play a pivotal role in the interaction between microbes and their host [81,82]. Furthermore, mounting evidence has demonstrated the role of microbial metabolites in modulating metabolic diseases such as obesity, by mediating epigenetic modification [22,83]. Here, we summarize the relationship between metabolites produced by the gut microbiota and epigenetic modifications in obesity (Figure 1).

### 4.1. Short-Chain Fatty Acids (SCFAs)

Short-chain fatty acids (SCFAs), including acetate, propionate, and butyrate, are generated by gut microbiota such as *Lactobacillus* and *Eubacterium* through the fermentation of indigestible polysaccharides, such as dietary fiber [84,85]. The role of SCFAs is now acknowledged to encompass the epigenetic control of gene expression. For example, butyrate, a widely recognized histone deacetylase inhibitor with known epigenetic activity, impacts histone deacetylases and methyl CpG-binding proteins, thus potentially influencing DNA methylation [86]. Additionally, acetate has been shown to increase the acetylation levels of H3K9, H3K27, and H3K56 in the promoter regions, thereby activating the expression of lipid-synthesis genes, such as acetyl-CoA carboxylases alpha (*ACACA*) and fatty acid synthase (*FASN*), and influencing lipid synthesis [87].

Free fatty acid receptors (*FFARs*) are highly expressed in host adipose tissue [88]. It has been found that SCFAs can promote leptin secretion in adipocytes by activating FFARs, thereby regulating appetite and improving obesity [89]. In human type 2 diabetes patients, it has been observed that a lower abundance of the major butyrate producer, *F. prausnitzii*, leads to higher methylation in the CpG sites in the promoter region of the free fatty acid receptor *FFAR3* gene (*p* = 0.003) [63]. Additionally, Guo et al. [90] found that propionate enrichment in the obesity-prone population induces specific DNA methylation patterns in the *DAB adaptor protein 1 (DAB1)* promoter, a diabetes target gene (*p* < 0.05). This study highlights the potential mechanism by which alterations in epigenetic mechanisms induced by microbial metabolites may contribute to the susceptibility of obesity and other metabolic disorders, providing new therapeutic perspectives for the treatment of these diseases. Lu et al. [91] discovered that SCFAs decreased the expression of DNA methyltransferases (DNMT1, 3a, 3b) in high-fat-diet-induced obese mice, resulting in a reduction in CpG methylation in promoters of the leptin promoter (*p* < 0.05), thereby suppressing the obesity-related elevated leptin expression. The researchers hypothesized that the potential mechanism underlying the modulation of leptin’s epigenetic modifications by SCFAs may involve the inhibitory effect of SCFAs on HDACs, subsequently affecting the activities of HDACs and methyl CpG-binding proteins. Hence, it is plausible that epigenetic regulation plays a role in the advantageous effects of SCFAs on host metabolism. These findings may provide a new perspective for the treatment of obesity and other metabolic diseases.

### 4.2. Folate

Folate is an essential vitamin in the human diet and can be produced by bacteria such as *Bifidobacterium*, *Lactobacillus*, and *Bacillus subtilis*. Folate, serving as a methyl donor (MD), plays a vital role in methylation reaction, which encompasses a comprehensive network of interconnected metabolic pathways [92,93]. Inadequate/excessive folate intake may lead to abnormal expression of obesity-related genes and more severe obesity [94], thus providing insights into new perspectives for identifying the relationship between the gut microbiota, folate, epigenetic modulation, and obesity. A study showed that folic acid supplementation decreased body weight and reduced the level of DNA methylation at the DMRs of adenylate cyclase 3 (*Adcy3*) and Rap guanine nucleotide exchange factor 4 (*Rapgef4*) in HFD mice (*p* < 0.05) [95]. In addition, after consuming folate, obese women showed higher levels of DNA methylation compared to normal-weight women (*p* < 0.05) [96]. Researchers speculated folate influences DNA methylation status through its involvement in one-carbon metabolism, thereby mediating metabolic regulation in obesity [96]. Given the crucial role of maternal folate supplementation in fetal development and metabolism, a study conducted by Pauwels S et al. [97] found that there is a positive correlation between the duration of maternal folate supplementation before conception and the average CpG methylation level of the leptin gene (*p* = 0.024). Meanwhile, Haggarty et al. [98] observed higher methylation levels of the leptin gene in umbilical cord blood after folate supplementation initiated after 12 weeks of gestation (*p* = 0.044). Thus, maternal methyl-group donor intake during pregnancy can influence offspring DNA methylation in metabolism-related genes. Another recent study found that the prenatal supplementation of high-dose folate in obese pregnant mice resulted in disrupted lipid metabolism in the offspring, with significantly increased DNA methylation levels of CpG sites within the promoter of adipose triglyceride lipase (*ATGL*) in the liver and lipoprotein lipase (*LPL*) in adipose tissue (*p* < 0.05) [99]. In addition, dietary protein restriction and folate supplementation during pregnancy in rats significantly reduced the methylation level of PPAR genes (*p* < 0.001) in the offspring’s liver, thereby improving the risk of obesity and metabolic diseases in the offspring [100]. Collectively, these data support an association between folate, epigenetics, and obesity development, providing a potential role of the gut microbiota in mediating obesity by modulating folate production.

### 4.3. Choline

Choline, as a semi-essential nutrient for the human body, is found in various foods. One of its primary functions is to provide one-carbon units for the synthesis of DNA methylation donors [101]. Bacteria such as Faecalibacterium and Bacteroides can metabolize choline into trimethylamine (TMA), which regulates lipid metabolism and improves obesity [102]. Romano et al. [103] investigated the impact of the interaction between gut microbiota-mediated choline metabolism and DNA methylation on obesity-related diseases by engineering a microbial community lacking a single choline-utilizing enzyme. They found that mice colonized with the choline-consuming bacteria exhibited lower DNA methylation and increased inguinal fat accumulation than those mice colonized with bacteria that are unable to consume choline when fed with HFD (*p* < 0.01). The authors suspected that a bacterial choline metabolism decreased methyl donors and lowered global DNA methylation in the host, ultimately exacerbating HFD-induced metabolic disorders. Moreover, when compared to mothers who do not harbor choline-consuming bacteria in their bodies, the brains of offspring from mothers with such bacterial colonization exhibit lower levels of DNA methylation.The results suggest that gut microbiota-mediated choline metabolism can modulate the development of obesity by altering the DNA methylation and also influence the DNA methylation profiles in offspring.

### 4.4. Polyphenols

Polyphenols are a class of naturally occurring compounds with widespread distribution and diverse biological activities [104]. Increasing research has found that polyphenols are primarily metabolized by the colonic microbiota, forming more bioactive metabolites than those consumed in food, which affect the composition of the intestinal microbiota and metabolites [105,106]. Moreover, the polyphenol metabolites primarily alter cellular functions by regulating miRNA levels, thus modulating the occurrence of obesity [107]. This provides a new perspective on the role of polyphenols in preventing HFD-induced obesity. Zhen Wang et al. found that supplementation with polyphenols can regulate the composition and abundance of the intestinal microbiota in obese mice, leading to an increase in SCFAs. It also inhibits the expression of several obesity-related microRNAs in the inguinal or epididymal white adipose tissue of obese mice, such as miR-200c-3p and miR-125a-5p (*p* < 0.05) [108]. *Akkermansia muciniphila* has been proven to be a probiotic that regulates obesity [109]. After supplementation with polyphenols in mice, the abundance of *Akkermansia muciniphila* and the expression of miR-30d both increased [110]. Polyphenols and their microbial metabolites may mediate the host’s metabolic disorders by regulating intestinal miRNAs.

Together, these groundbreaking insights significantly contribute to the advancement and comprehension of the connections between gut-microbiota-derived metabolites and the epigenetic status associated with obesity. Based on these relevant data, efforts to improve bacterial populations and induce beneficial epigenetic changes may offer a new direction for the effective prevention of obesity and associated clinical manifestations.

## 5. Clinical Relevance in Obesity

The understanding of the significant role played by the gut microbiota and epigenetics in energy metabolism, low-grade inflammation, and maternal inheritance has paved the way for innovative nutritional therapeutic strategies to address obesity [111,112]. These therapeutic approaches involve microbiota-targeted interventions, such as the use of beneficial microbiota (probiotics) or the promotion of microbial growth (e.g., prebiotics), which can influence the intricate relationship between the microbiota and epigenetics [44,113].

### 5.1. Probiotics

Probiotics are viable microorganisms that, when administered at therapeutic doses, can provide health benefits to the host by influencing gut flora [114,115]. Moreover, probiotic supplements can induce epigenetic modifications that may alter the expression of genes involved in lipid metabolism, thereby reducing the risk of obesity [116,117].

Given the role of the gut microbiota and epigenetic modifications in metabolic health, it is believed that probiotics can exhibit metabolic effects by interacting with the host’s epigenetic mechanisms. The following focus is on investigating the effects of different probiotics on the epigenetic modifications of obesity-related genes. As stated in Table 3, supplementation with probiotics inhibited the high methylation of *H3K27me3* at the mitochondrial transcription factor A (*TFAM*) promoter in obese mice (*p* < 0.001), thereby improving obesity-induced metabolic osteoporosis [118]. Additionally, supplementation of *L. rhamnosus GG* (*LGG*) and *B. lactis* to pregnant women has been shown to decrease the DNA methylation of the fat mass and obesity associated gene (*FTO*) and melanocortin-4 receptor gene (*MC4R*) in both women and their infants (*p* < 0.05) [69]. The significance of these findings lies in the involvement of probiotics in regulating the DNA methylation patterns of genes associated with energy metabolism. Moreover, *L. rhamnosus* alleviates lipid metabolism disorders and weight gain in obese mice by increasing the expression of miR-155-5p, miR-155-5p, and miR-26a-5p in the liver (*p* < 0.05) [119]. However, the clinical outcomes of probiotics in alleviating obesity and other metabolic diseases through epigenetic mediation are variable. Although previous findings have indicated that probiotic supplementation could improve the expression of *miR-26a-5p* in obese mice (*p* < 0.05), significant effects on its expression were not observed in human clinical trials [120]. Therefore, the clinical effects of probiotics may depend on specific species and strains used, and further clinical research is needed to ascertain the dosage, treatment duration, and long-term effects of different strains.

### 5.2. Prebiotics

Prebiotics refer to fermentable substances that are selectively metabolized by the microbiota, leading to specific changes in its composition and/or activity, ultimately providing advantages to the host [128,129]. Common examples of prebiotics include inulin, fructo-oligosaccharides (FOS), and mannan-oligosaccharides (MOS) [130]. Maternal inulin supplementation improved glucose metabolism impairment and insulin resistance by activating wnt family member 5a (*Wnt 5a*) methylation and inhibiting phosphatidylinositol-4,5-bisphosphate 3-kinase catalytic subunit alpha (*PiK3CA*) methylation in offspring livers exposed to a maternal HFD (*p* < 0.01) [131]. Additionally, inulin intervention significantly reduced body weight, waist circumference, and body mass index by reducing the methylation levels of uric acid and four CpG sites in the promoter region of the *insulin* gene in patients with type 2 diabetes [124]. This finding highlights the critical role of inulin, considered as a prebiotic, in alleviating obesity and its related metabolic disorders via regulating the methylation process. In addition, experimental data from both clinical and preclinical studies have primarily focused on the role of probiotics and prebiotics in mediating epigenetic regulation and influencing metabolic mechanisms, as summarized in Table 3.

### 5.3. Fecal Microbiota Transplant

Fecal microbiota transplantation (FMT) is a therapeutic intervention to restore host health by enhancing the diversity and functionality of the gut microbiota [132,133,134]. Human randomized trials have provided evidence that FMT, transferring fecal material from healthy donors to patients with metabolic syndrome, leads to increased levels of SCFA-producing bacteria, notable changes in plasma metabolites involved in lipid metabolism, and reduced methylation levels of the actin-filament-associated protein 1 (*AFAP1*) promoter [135]. Conversely, in a mouse model, FMT from obese-susceptible donor mice resulted in exacerbated insulin resistance and higher levels of DNA methylation at two specific CpG sites in the colon tissue (*p* < 0.05) [90]. FMT has also shown promise in improving non-alcoholic fatty liver disease, depression, and other conditions through epigenetic modification, offering a potential new avenue for treating human-related diseases [136,137]. However, given the complexity of the human gut microbiota ecosystem, challenges related to engraftment, such as microbiota resilience and host environmental filtering, need to be considered in future FMT studies [138,139,140].

Consequently, probiotics, prebiotics, and FMT can serve as a bridge between the gut microbiota and host metabolism, altering health outcomes by modulating, at least partially, epigenetic mechanisms.

## 6. Conclusions

Obesity has emerged as a considerable global health hazard. The gut microbiota plays a pivotal role in human metabolism, serving as a major contributor to overall health outcomes. Potential microbial metabolites can also interact with cells through systemic circulation, acting as one of the critical environmental factors influencing the epigenome. Obesity-induced metabolic dysregulation and disruptions in gut microbiota composition may lead to imbalances in key metabolites, subsequently affecting epigenetic pathways and altering gene expression. Consequently, increasing attention is being paid to the intricate interplay between the gut microbiota and epigenetic modifications in the context of metabolic diseases. In this review, we aim to summarize the current research on this interaction. Several studies have demonstrated that the gut microbiota can directly modulate the epigenome, as well as produce epigenetic substrates and enzyme co-factors. Alternatively, they can target proteins or genetic regulatory regions through microbe-derived metabolites to achieve specific epigenetic modifications, thereby altering the epigenetic programming of metabolic pathways. In summary, the integration of epigenetic mechanisms and gut microbiota data showcases how environmental factors can lead to obesity, providing novel interventional strategies for the treatment of metabolic diseases. Indeed, this review summarizes the studies on probiotics, prebiotics, and other modulators that adjust gut microbiota composition and influence epigenetic mechanisms, thus contributing to obesity management. These studies may pave the way for clinical applications of the gut microbiota as a therapeutic target in the prevention and treatment of obesity. Moreover, with the growing demand for dietary supplements and nutraceuticals, these products offer effective and safe support for the medical nutrition therapy of obesity. However, to date, our understanding of the epigenetic mechanisms of the gut microbiota in obesity is primarily derived from rodent models, lacking validation from human clinical trials. Additionally, it is imperative to further elucidate the precise role of specific gut microbiota strains in regulating the epigenome during obesity. Therefore, further clinical investigations into the intricate interplay among gut microbiota, epigenetic modifications, and obesity are crucial for human health and the management of metabolic diseases.

In conclusion, these pioneering insights contribute significantly to our understanding of the interactions between gut microbiota composition and epigenetic modifications in metabolic regulation. They are vital for designing and implementing novel personalized care, improving drug selection, and preventing and managing obesity and its comorbidities.

## Figures and Tables

**Figure 1 ijms-25-07175-f001:**
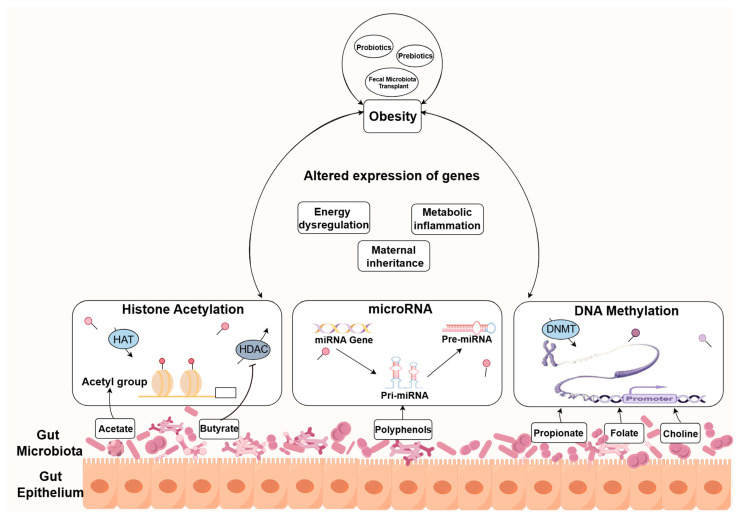
The interplay between the gut microbiota, epigenetic modifications, and obesity. The gut microbiota serves as a source of epigenetic factors, producing substrates or co-factors that modulate the epigenetic enzymes involved in energy metabolism, metabolic inflammation, and maternal inheritance-related gene epigenetic modifications, ultimately influencing the development of obesity-related diseases. HATs: histone acetyl transferases; HDAC: histone deacetylases; DNMTs: DNA methyltransferases.

**Table 1 ijms-25-07175-t001:** The gut microbiota in obesity is associated with epigenetics in the context of low-grade inflammation.

Study Design	Method	Changes in the Gut Microbiota	Effect on Epigenetic Modulation	Effect on the Host Relating to Obesity	Ref.
The gut microbiota of mice consists of two groups: those fed a high-fat diet and those fed a normal diet (*n* = 5).	16S rRNA gene sequencing of stool samples.	Decreased diversity of the gut microbiota and a reduction in ethanolamine-metabolizing bacteria (*p* < 0.001).	Elevated levels of ethanolamine increase the expression of miR-101a-3p (*p* < 0.001).	Reducing bacteria responsible for metabolizing ethanolamine, preserving intestinal-barrier integrity, and preventing an increase in intestinal permeability.	[62]
A comparison was made between the gut microbiota of high-fat/high-sucrose male rats and control rats (*n* = 12).	16S rRNA gene sequencing of stool samples.	The abundance of *Faecalibaculum* and *Bifidobacterium* significantly decreases (*p* < 0.01).	Dysregulation of bacteria involved in short-chain fatty acid production is associated with the methylation levels of the promoter of free fatty acid receptors (*p* = 0.031).	Reduction in *Bacteroides* and *Bifidobacterium* hampers the production of short-chain fatty acids, decreases the population of Treg cells, and disrupts intestinal metabolic homeostasis.	[63,64]
A group of diabetic patients were divided into inulin-fructan and placebo treatment groups for 6 weeks to compare their gut microbiota composition (*n* = 25).	16S rRNA gene sequencing of stool samples.	Microbial community diversity is lower in patients with type 2 diabetes, with a lower abundance of *Bifidobacterium (p* = 0.045).	*Bifidobacterium* is associated with the inhibition of the histone acetylation of inflammatory factors interleukin-17 gene (*IL-17)* and interleukin-23 gene (*IL-23)* (*p* < 0.05).	Inhibition of adipocyte cytokine expression is accompanied by dyslipidemia, leading to low-grade chronic inflammation.	[65,66]

**Table 2 ijms-25-07175-t002:** The gut microbiota in obesity is associated with the role of epigenetics in maternal inheritance.

Study Design	Method	Changes in the Gut Microbiota in Maternalism	Effect on Epigenetic Modulation in Offspring	Effect on the Host Relating to Obesity in Maternalism	Ref.
A comparison of the gut microbiota in pregnant women (*n* = 10).	16S rRNA gene sequencing of stool samples.	The major bacterial taxa in late pregnancy are *Firmicutes*.	A link exists between changes in the methylation of type 2 diabetes-associated genes in fetuses and the microbiota components in mothers during pregnancy (*p* < 0.05).	Dysbiosis of the *Firmicutes* phylum may lead to increased energy intake, resulting in the accumulation of fat.	[71,76]
A comparison of the gut microbiota in the offspring of women with gestational diabetes (*n* = 10) and the offspring of women with normal blood sugar levels (*n* = 19).	16S rRNA gene sequencing of stool samples.	The relative abundance of *Escherichia coli* and *Bacteroides* is significantly higher (*p* < 0.001).	*Escherichia coli* is associated with the expression of long non-coding RNA (lncRNA) that participates in inflammation signaling (*p* < 0.05).	Microbes associated with energy metabolism pathways exhibit an increased abundance, leading to an increase in obesity prevalence.	[77,78]
Two groups of mother mice were fed with a high-fat diet and a normal diet, respectively (*n* = 5).	16S rRNA gene sequencing of cecal contents.	Dysbiosis of the gut microbiota with decreased α-diversity.	The methylation patterns of genes associated with liver fibrosis and lipid accumulation, specifically the DMRs, are altered in the offspring (*p* < 0.01).	Excessive accumulation of fat in liver cells leads to the development of fatty liver.	[79,80]

**Table 3 ijms-25-07175-t003:** Probiotic and prebiotic studies for metabolic diseases.

Study Design	Method	Alterations in the Gut Microbiota	Effects on Epigenetic Modulation	Effects on Obesity	Ref.
Probiotics					
Supplementing probiotic capsules containing *Lactobacillus rhamnosus GG* and *Bifidobacterium lactis Bb12* in pregnant women (*n* = 7).	16S rRNA gene sequencing of stool samples.	Enhancing the abundance of beneficial bacteria in the host (*p* < 0.05).	Increasing abundance is associated with an increase in the methylation activity of the *IGFBP1* promoter (*p* < 0.001).	Improving glucose metabolism and obesity.	[72]
Supplementation of *Lactobacillus* in mice induced by a high-fat diet (*n* = 4).	16S rRNA gene sequencing of stool samples.	Increasing *L.* spp. and *B. animalis* (*p* < 0.01).	The crosstalk between H3K79me2 and H3K27me3 histone modifications alters the expression of forkhead box O1 (*FOXO1*) (*p* < 0.001).	Improving insulin resistance.	[121]
Obese mother mice were supplemented with a mixture of probiotics (VSL#3) (*n* = 6).	16S rRNA gene sequencing of stool samples.	The increasing diversity of the gut microbiota suggests an expansion of the proportion of *Clostridium species* involved in tryptophan metabolism (*p* < 0.0001).	Increasing the activity of histone demethylase Kdm6b (*p* = 0.0001).	Reducing intestinal permeability and inflammation.	[118]
Supplementation of *Lactobacillus rhamnosus, LR*, to diabetic mice (*n* = 8).	16S rRNA gene sequencing of stool samples.	Increasing the abundance of *Roseburia* and *Lactococcus* among others (*p* < 0.05).	Reversing the expression of miR-155-5p, miR-26a-5p, and other liver-metabolism-related H3K27me3 histone modifications caused by obesity (*p* < 0.05).	Decreasing blood glucose and triglyceride levels and regulation of gluconeogenesis.	[119,122]
Prebiotics					
Supplementation of type 2 diabetes with inulin (*n* = 4).	16S rRNA gene sequencing of stool samples.	Decreasing *Bacteroides*, *Ruminococccus*, and increasing *Alistipes* (*p* = 0.045).	Decreasing the methylation levels of the insulin (*INS*) gene (*p* = 0.0001).	Reducing blood glucose levels in diabetic patients.	[123,124]
Supplementing pregnant mice fed with a high-fat diet with inulin (*n* = 6).	16S rRNA gene sequencing of stool samples.	Increasing the abundance of *Bifidobacterium* in the intestines of their offspring (*p* = 0.049).	Inhibited the methylation of *Lepr* in the hypothalamus of offspring (*p* < 0.05).	Modifying offspring lipid metabolism	[125,126]
Supplementing maternal mice on a high-fat diet with oligofructose (*n* = 15).	16S rRNA gene sequencing of stool samples of the offspring.	Increasing the abundance of *Bifidobacterium* in the offspring (*p* < 0.05).	Reducing the levels of miR-26a and miR-27a in the breast milk of the high-fat-diet-fed mother mice (*p* < 0.05).	Contributing to improved glucose tolerance in the offspring and reduced the likelihood of insulin resistance and hepatic steatosis in the offspring.	[123,127]

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
