# Peer review of "The Gut Microbial Regulation of Epigenetic Modification from a Metabolic Perspective"

_ijms, 2024, doi:10.3390/ijms25137175_

Round 1
Reviewer 1 Report
Comments and Suggestions for Authors
Dear Sirs,
The authors of this manuscript discuss the association of gut microbiota and its metabolites with epigenetic mechanisms in obesity, elucidating the impact of gut microbiota as an epigenetic modifier in obesity-related conditions. While the topic of this review is interesting, I recommend the following changes:
1.Lines 16-18: I propose the removal of the phrase “We conducted searches on the PubMed, Web of Science, and Google databases using keywords such as 'gut microbiota,' 'gut microbiome,' 'epigenetics,' and 'obesity' to identify relevant articles published from 2006 to January 2024.” Instead, the authors may incorporate a section called Literature Search Methodology right after the Introduction as section 2. Literature Search Methodology. In this section, the authors should detail their literature search methodology, explaining the criteria and rationale for selecting certain papers and excluding others.
2.Lines 28-30: Please rephrase as follows: The global prevalence of obesity continues to rise at an alarming rate. Recent statistics reveal that over 200 million adults are afflicted with obesity or other metabolic diseases, accounting for approximately 30% of the world's population.
3.Lines 43-45: Please rephrase as follows: Therefore, a comprehensive understanding of the underlying epigenetic mechanisms involved in the development of obesity is crucial. Such knowledge has the potential to pave the way for promising therapeutic strategies to combat obesity.
4.Line 94: Please add the abbreviation for non-coding RNAs.
5.Section 2: The authors may enrich their manuscript with additional evidence related to maternity. This review (PMID: 38785533) could provide significant guidance on the subject.
6.Section 3: Are there any evidence regarding the plant-based diets, polyphenols, MUFAs, and PUFAs?
7.Line 304: Please remove the term syndrome.
8.The reference list should be formatted according to the journal’s specific instructions and guidelines.
9.Extensive editing for English language is required.
Comments on the Quality of English LanguageExtensive editing for English language is required.
Author Response
Dear reviewer,
We are deeply grateful for the opportunity to refine our paper, titled "The gut microbial regulation of epigenetic modification from a metabolic perspective" (ID: ijms-3057427). We sincerely appreciate the reviewers' and editor's insightful comments and recommendations, which have been immensely valuable. We have carefully revised our manuscript, incorporating the feedback received, and we have highlighted all the changes in red to facilitate easy review. We hope that these revisions will further strengthen our paper and enhance its chances of acceptance. Below, we have addressed the revisions in a point-by-point manner.
Reviewers’ Comments to Author:
Reviewer#1:
Dear Sirs,
The authors of this manuscript discuss the association of gut microbiota and its metabolites with epigenetic mechanisms in obesity, elucidating the impact of gut microbiota as an epigenetic modifier in obesity-related conditions. While the topic of this review is interesting, I recommend the following changes
1.Lines 16-18: I propose the removal of the phrase “We conducted searches on the PubMed, Web of Science, and Google databases using keywords such as 'gut microbiota,' 'gut microbiome,' 'epigenetics,' and 'obesity' to identify relevant articles published from 2006 to January 2024.” Instead, the authors may incorporate a section called Literature Search Methodology right after the Introduction as section 2. Literature Search Methodology. In this section, the authors should detail their literature search methodology, explaining the criteria and rationale for selecting certain papers and excluding others.
Response: We thank the reviewer for this thought-provoking suggestion. Based on the reviewer's opinion, We have deleted this part and re-added the second section, “Literature Search Methodology”.
2.Lines 28-30: Please rephrase as follows: The global prevalence of obesity continues to rise at an alarming rate. Recent statistics reveal that over 200 million adults are afflicted with obesity or other metabolic diseases, accounting for approximately 30% of the world's population.
Response: We sincerely thank the reviewer for their careful reading. Based on the reviewer's comments, We have revised this section and re-added references, such as PMID: 26045323.
3.Lines 43-45: Please rephrase as follows: Therefore, a comprehensive understanding of the underlying epigenetic mechanisms involved in the development of obesity is crucial. Such knowledge has the potential to pave the way for promising therapeutic strategies to combat obesity.
Response: We appreciate the thoughtful review and constructive feedback provided by the reviewer. We have revised this section as requested.
4.Line 94: Please add the abbreviation for non-coding RNAs.
Response: We sincerely thank the reviewer for their careful reading. We have added the abbreviation of non-coding RNAs (ncRNAs) at this point.
5. Section 2: The authors may enrich their manuscript with additional evidence related to maternity. This review (PMID: 38785533) could provide significant guidance on the subject.
Response: We are extremely grateful for your professional advice. We have carefully reviewed the comprehensive review and supplemented the genetic-related content in paragraphs 207-210 and lines 292-295.
6. Section 3: Are there any evidence regarding the plant-based diets, polyphenols, MUFAs, and PUFAs?
Response: We are deeply grateful for your careful reading and professional evaluation. Based on the reviewer's suggestions, we have focused on reviewing the literature related to polyphenols and supplemented Section 4.4 in the manuscript accordingly.
7.Line 304: Please remove the term syndrome.
Response: We appreciate the thoughtful review and constructive feedback provided by the reviewer. We have removed the term "syndrome" from the document.
8.The reference list should be formatted according to the journal’s specific instructions and guidelines.
Response: Thank you for your suggestion. We have modified the reference format according to the journal's specific instructions.
9.Extensive editing for English language is required.
Response: We are very grateful for your valuable suggestions. We have made every effort to improve the manuscript and have made some revisions accordingly. These changes will not affect the content and structure of the paper. Here, we attach the language polishing certificate as a supporting document. We sincerely appreciate the reviewer's advice and hope that the corrections will be approved.
We are profoundly grateful for your thoughtful feedback on our paper. We eagerly await your further insights and hope to hear from you soon.
With best wishes,
Yours sincerely,
Xingtong Lin
2024-6-24

Reviewer 2 Report
Comments and Suggestions for Authors
The manuscript presented by Lin et al. provides a coherent review of studies focusing in the microbial regulation of epigenetic modification in the context of obesity. The axis microbiome modification-epigenetic modification is well explored, allowing the perception of the existence of a causal effect between changes in species of the microbiome and distinct outcomes in host epigenetics. A relation is also presented when looking at variety of metabolites produced by the microbiome and epigenetic alterations induced in the host.
Major comment:
i) The figure 1, prepared by the authors to schematically show the “Interplay between gut microbiota, epigenetic modifications, and obesity”, does not pay a good tribute to the overall quality of the manuscript. A more exhaustive and comprehensive schematics is advisable.
ii) The authors present tables with clinical trials focusing topics covered in their revision manuscript but do not report to their major outcomes in the text of the manuscript. They should be discussed in some detail in the manuscript text itself and not slowly presented as Tables.
iii) The conclusion section needs to be significantly improved. Since there is no discussion section in the manuscript, this conclusion section can accommodate a more integrative concept of all aspects presented in the review.
Minor comments/suggestions:
i) The first sentence in the introduction need’s correction. If prevalence is on the order of 30% than a number of 200 million adults does not represent such a percentage. Also, the reference chosen to justify such a sentence in the manuscript ([1]) is most certainly not the more suitable and should be altered.
ii) Page 1 lines 31-32: if the authors refer including they do not need to finish the sentence by “and more”.
iii) Page 3, lines 136-137: the sentence “Stearoyl-CoA desaturase 5 (SCD5), a key regulator of energy metabolism, is a fatty acid desaturase” is misplaced. The definition, if made, needs to be inserted in the following sentence in the manuscript.
iv) Page 7, line 233: there is repetition of “free fatty acid receptor” with no need.
v) Page 7, line 235: FFAR means already receptor thus “FFAR receptors” should either be substituted by FFA receptors or by FFAR.
vi) Page 7, lines 271-272: “higher methylation levels of leptin”; what should be mentioned is higher methylation levels of leptin gene. It’s not the hormone itself that gets methylated.
Author Response
Dear reviewer,
Thank you very much for allowing us to revise our manuscript entitled "The gut microbial regulation of epigenetic modification from a metabolic perspective" (ID: ijms-3057427), we would like to thank the reviewers and the editor for the positive and constructive comments and suggestions. We have substantially revised our manuscript after reading the comments provided by the reviewers and found these comments to be very helpful. In the revised manuscript, all the changes are highlighted in red for easy inspection. We hope this revision can make our paper more acceptable. The revisions were addressed point by point below.
Reviewers’ Comments to Author:
Reviewer#2:
The manuscript presented by Lin et al. provides a coherent review of studies focusing in the microbial regulation of epigenetic modification in the context of obesity. The axis microbiome modification-epigenetic modification is well explored, allowing the perception of the existence of a causal effect between changes in species of the microbiome and distinct outcomes in host epigenetics. A relation is also presented when looking at variety of metabolites produced by the microbiome and epigenetic alterations induced in the host.
Major comment:
- The figure 1, prepared by the authors to schematically show the “Interplay between gut microbiota, epigenetic modifications, and obesity”, does not pay a good tribute to the overall quality of the manuscript. A more exhaustive and comprehensive schematics is advisable.
Response: We greatly appreciate your professional comments on our articles. We apologize for any confusion caused by this section. Based on your suggestions, we have made corrections and additions to the diagram. Firstly, we have added the concept of "miRNA," which ensures that the diagram now covers all the epigenetic mechanisms mentioned in this article. Secondly, we have labeled "gut microbiota." This indicates that the interaction between gut microbiota and their metabolites with epigenetic modification mechanisms can influence changes in epigenetic regulation of genes related to obesity pathways, thus mediating the occurrence of obesity diseases. Finally, we have supplemented the clinical treatment approaches mentioned in Section 5.1, making the diagram more consistent with the logic of the article.
- The authors present tables with clinical trials focusing topics covered in their revision manuscript but do not report to their major outcomes in the text of the manuscript. They should be discussed in some detail in the manuscript text itself and not slowly presented as Tables.
Response:Thank you for your suggestions. In this manuscript, we have primarily included three tables, located in sections 3.2, 3.3, and Part 5. The main purpose of inserting these tables is to supplement the related content with other studies, therefore, they have not been described in detail in the text. I have highlighted the studies mentioned in the tables with red text in paragraphs 144 and 155.
- The conclusion section needs to be significantly improved. Since there is no discussion section in the manuscript, this conclusion section can accommodate a more integrative concept of all aspects presented in the review.
Response: We greatly appreciate your professional advice, and based on your suggestions, we have made significant revisions to the "Conclusion" section. In the revised summary, we have provided a more comprehensive overview and evaluation of all the concepts mentioned in the manuscript, and have also put forward some suggestions for future developments in this area.
Minor comments/suggestions:
- The first sentence in the introduction need’s correction. If prevalence is on the order of 30% than a number of 200 million adults does not represent such a percentage. Also, the reference chosen to justify such a sentence in the manuscript ([1]) is most certainly not the more suitable and should be altered.
Response: Thank you very much for your professional comments on our manuscript. We apologize for any confusion the manuscript may have caused. Based on the reviewer's suggestions, we have revised this section and added new references, such as PMID: 26045323.
2. Page 1 lines 31-32: if the authors refer to including they do not need to finish the sentence by “and more”.
Response: We sincerely thank the reviewers for their careful reading. Based on the reviewer's feedback, we have revised the sentence and removed "and more".
3. Page 3, lines 136-137: the sentence “Stearoyl-CoA desaturase 5 (SCD5), a key regulator of energy metabolism, is a fatty acid desaturase” is misplaced. The definition, if made, needs to be inserted in the following sentence in the manuscript.
Response: We greatly appreciate your professional comments on our articles. We have deleted this sentence to make the logic of the manuscript clearer.
4. Page 7, line 233: there is repetition of “free fatty acid receptor” with no need.
Response: Thank you for your careful review. We have removed the term "free fatty acid receptor".
Page 7, line 235: FFAR means already receptor thus “FFAR receptors” should either be substituted by FFA receptors or by FFAR.
Response: We sincerely appreciate your valuable comments. We apologize for any confusion the manuscript may have caused. We have revised this part to "FFAR".
5. Page 7, lines 271-272: “higher methylation levels of leptin”; what should be mentioned is higher methylation levels of leptin gene. It’s not the hormone itself that gets methylated.
Response: Thank you very much for your careful review. We apologize for any confusion caused by the previous manuscript. We have revised this part to "higher methylation levels of leptin gene".
We would like to express our great appreciation to you for your comments on our paper. Looking forward to hearing from you soon.
With best wishes,
Yours sincerely,
Xingtong Lin
2024-6-24
Round 2
Reviewer 1 Report
Comments and Suggestions for Authors
The manuscript has been improved and it can be published.
Comments on the Quality of English LanguageMinor editing of English language required.
Reviewer 2 Report
Comments and Suggestions for Authors
Thanks for replying to my comments. I find your work very interesting.